# Biomimetic approach to the catalytic enantioselective synthesis of tetracyclic isochroman

Xiangfeng Lin[1,2], Xianghui Liu[1,2], Kai Wang[1,2], Qian Li[1,2], Yan Liu[1]✉ & Can Li[1]✉

Polyketide oligomers containing the structure of tetracyclic isochroman comprise a large class of natural products with diverse activity. However, a general and stereoselective method towards the rapid construction of this structure remains challenging due to the inherent instability and complex stereochemistry of polyketide. By mimicking the biosynthetic pathway of this structurally diverse set of natural products, we herein develop an asymmetric hetero-Diels–Alder reaction of in-situ generated isochromene and *ortho*-quinonemethide. A broad range of tetracyclic isochroman frameworks are prepared in good yields and excellent stereoinduction (up to 95% *ee*) from readily available *α*-propargyl benzyl alcohols and 2-(hydroxylmethyl) phenols under mild conditions. This direct enantioselective cascade reaction is achieved by a Au(I)/chiral Sc(III) bimetallic catalytic system. Experimental studies indicate that the key hetero-Diels-Alder reaction involves a stepwise pathway, and the steric hindrance between in-situ generated isochromene and *t*-Bu group of Sc(III)/*N,N′*-dioxide complex is responsible for the enantioselectivity in the hetero-Diels–Alder reaction step.

[1] State Key Laboratory of Catalysis, Dalian Institute of Chemical Physics, Chinese Academy of Sciences, Dalian, China. [2] University of Chinese Academy of Sciences, Beijing, China. ✉email: yanliu503@dicp.ac.cn; canli@dicp.ac.cn

Tetracyclic isochromans, a type of polyketide oligomers[1–3], are ubiquitously present in numerous natural products and bioactive molecules. For instance, Chaetophenol **D** was induced from chemically mediated epigenetic manipulation of fungal gene expression, exhibiting excellent *anti*-adenovirus activity[4]. Indigotide **C** and its isomer were biosynthesized by cultivation of *Cordyceps indigotica*, an entomopathogenic fungus, in the presence of suberoyl bis-hydroxamic acid, exhibiting good antibacterial activity[5] (see Fig. 1a). Thus, the development of a general and stereoselective method towards a rapid construction is highly desirable. However, to our knowledge, such an effient methodology has not yet been established so far[6], probably due to the inherent instability of tetracyclic isochroman polyketide and the difficulties in controlling the complex stereochemistry.

As an important intermediate, *ortho*-quinone methide (*o*-QM) has caught much attention in the field of asymmetric catalysis[7–26]. Moreover, this highly reactive specie was proposed to be the diene intermediate in the biosynthesis of several polyketide natural products via hetero-Diels–Alder (HDA) reaction. Asai and Oshima speculated a biosynthetic route of Indigotide **C** involving isochromene and *o*-QM intermediates[5]. The same group developed a biosynthetic approach for the construction of a pseudo-natural fungal polyketide **A** by using **1** as substrate in 2015[27] (see Fig. 1b). The key of this method also lies in a HDA reaction of isochromene and *o*-QM which were heterologously biosynthesized by introducing the NR-PKS-encoding gene into *Aspergillus oryzae*. Inspired by this possible biosynthetic route of poliketide oligomer, we decided to exploit an artificial asymmetric catalytic system from the similar intermediates to build this complex structure. To the best of our knowledge, it is worth noting that there has been no report of biomimetic asymmetric catalytic HDA reaction involving *o*-QMs intermediates. The development of biomimetic HDA route for the synthesis of core structure of these natural products is of great significance to not only understand biosynthetic HDA reactions involving *o*-QM

intermediates, but also expand the limitation of substrates catalyzed by Diels–Alderase enzymes[28–34]. The establishment of the enantioselective biomimetic polyketide oligomer synthesis requires the methods for the generation of isochromene and *o*-QM intermediates and a catalyst capable of promoting the stereoselective HDA reaction. It is known that Au(I) complexes have been used in the cascade reactions[35–39] and could promote the intramolecular cyclization of benzyl alcohol-functionalized alkynes to isochromene intermediate[40,41], and the *o*-QM intermidiate could be generated from the proper precursor in the presence of either acid or base.

Recently, Au/Lewis acid bimetallic catalysis has been exploited to be a strategy to construct aminals and ketals via *5-exo-dig* cyclization of alkynyl amides or alcohols and relay cycloaddition reactions with electrophiles[42–49]. Moreover, Feng's group developed the first highly efficient asymmetric bimetallic Au(I)/chiral Ni(II) catalytic cascade reaction of $\alpha$, $\beta$-unsaturated $\gamma$-keto esters with alkynyl amides and alcohols for the synthesis of spiroketals and spiroaminals in 2016[50]. Kang and coworkers realized the same reaction by employing a Au(I)/chiral Rh(III) catalytic system two years later[51]. Inspired by these elegant works, we envisioned that a combination of a Au(I) catalyst and a chiral lewis acid catalyst might enable the HDA reaction of in-situ generated isochromene and *o*-QM to deliver the optically active tetracyclic isochromans, despite of that the acceptors of reported asymmetric Au/Lewis acid bimetallic catalysis were limited to $\alpha$, $\beta$-unsaturated $\gamma$-keto esters, and the most of known asymmetric cascade reactions of alkynyl alcohols delivered spiro products rather than fused products[50,52–61].

In this work, we found a highly efficient Au(I)/chiral Sc(III) bimetallic catalytic system which could achieve asymmetric HDA reaction of in-situ generated isochromene and *ortho*-quinonemethide (*o*-QM), and provided a diversity of tetracyclic isochromans in moderate to high yields and with high levels of diastereoselectivities and enantioselectivities (up to 95% *ee*) (see Fig. 1c).

**Fig. 1 Background of tetracyclic isochromans and biomimetic design. a** Natural polyketides containing tetracyclic isochromans. **b** The possible biosynthetic route of tetracyclic isochroman polyketide catalyzed by NR-PKS. **c** The bioinspired Au/Sc*catalyzed HDA reaction.

**Fig. 2 Optimization of the reaction.**

| Entry | Ligand | Solvent | Yield[a] | Ee/%(major)[b] | Ee/% (minor)[b] | D.r.[c] |
|---|---|---|---|---|---|---|
| 1 | - | DCM | 65 | N.D. | N.D. | 3/1 |
| 2 | L-PiPr$_2$ | DCM | 61 | 56 | 4 | 7/3 |
| 3 | L-PiMe$_3$ | DCM | 60 | 72 | 22 | 3/1 |
| 4 | L-PrPr$_2$ | DCM | 28 | 50 | 0 | 3/1 |
| 5 | L-PiPr$_3$ | DCM | 51 | 81 | - | 7/1 |
| 6 | L-PiAd | DCM | 61 | 30 | - | 7/1 |
| 7 | L-PiMe$_2$t-Bu | DCM | 66 | 66 | - | 9/1 |
| 8 | L-PiMe$_2$t-Bu | CHCl$_3$ | 60 | 60 | 0 | 3/1 |
| 9 | L-PiMe$_2$t-Bu | DCE | 42 | 70 | - | 7/1 |
| 10[d] | L-PiMe$_2$t-Bu | DCM | 65 | 91 | - | 9/1 |

All reactions were carried out on a 0.1 mmol scale with 1 eq precursor of o-QMs **2a**, 1.2 eq **3a**, 10 mol% of Sc(OTf)$_3$, 11 mol% of ligand and 5 mol% of JohnphosAu in DCM (1 mL) at room temperature. [a]Isolated yield. [b]Determined by chiral HPLC. [c]Determined by by crude $^1$H-NMR. [d]Reaction temperature: 6 °C.

## Results and discussion

Studies commenced by screening effective precursors of o-QMs and the corresponding catalysts. We selected α-propargyl benzyl alcohol **3a** as model substrate and JohnphosAu as the catalyst to generate isochromene intermediate. 2-(Hydroxylmethyl) phenol **2a** was chosen as the precursor of o-QM. Representative Lewis acids and Brønsted acids were investigated in the presence of 1.2 equivalent **3a** and JohnphosAu catalyst (see the Supporting Information, Table S2). Sc(OTf)$_3$ was found to be the only efficient catalyst for this reaction, affording the HDA product **4a** in 65% yield with 3/1 d.r. (Fig. 2, entry 1). Subsequently, a series of other noble metal complexes such as Pd(II), Ag(I) and Au(I) complex were evaluated to generate isochromene intermediate, but only (acetonitrile)Au-complex delivered the desire HDA products in good yield and no product was obtained by in-situ formed Au-complex (see the Supporting Information, table S2). Next, the screening of different types of chiral ligands for the purpose of achieving the asymmetric version of this reaction suggested that Feng ligands (N,N'-dioxide) showed unique activities in the asymmetric HDA reaction of in-situ generated isochromenes and o-QMs (Fig. 2, entries 2-7). The use of (S)-proline derived L-PrPr$_2$ resulted in lower diastereoselectivities and enantioselectivities than (S)-piperidine derived L-PiPr$_2$ (Fig. 2, entry 2 VS entry 4). More sterically hindered substituents at the para positions of aniline were beneficial to improve both diastereoselectivities and ee values (Fig. 2, entries 5 and 7 VS entries 2 and 3) and aniline derived L-PiMe$_2$t-Bu resulted in higher ee values than amantadine derived L-PiAd (Fig. 2, entry 6 VS entry 7). The solvent screening suggested that the catalytic reactions were available in chlorinated solvents and DCM gave the best yield and selectivity (Fig. 2, entries 7-9). Notably, after cooling to 6 °C, the ee value of the HDA product **4a** was increased

to 91% with little drop of yield (Fig. 2, entry 10). Therefore, the optimized conditions involved the use of L-PiMe$_2$t-Bu/Sc(OTf)$_3$ and JohnphosAu as catalysts in DCM at 6 °C (Fig. 2, entry 10).

Under optimized reaction conditions, the substrates scope of the reaction was examined. A series of aryl-substituted ortho-hydroxybenzyl alcohols **2a-i** were tested with **3a** as model α-propargyl benzyl alcohol (Fig. 3). Pleasingly, either electron-donating or electron-withdrawing substituents at the para- or meta-position of the benzene ring (**2a-g**) were well tolerated, affording the corresponding poliketide oligomers in moderate yields and with good to excellent diastereoselectivities and enantioselectivities. In particular, the reactions of o-QMs with 2-naphthyl and 4-biphenyl substituents successfully afforded the desired products (**4h,i**) in 76% and 60% yield with 90% and 84% ee respectively. However, ortho-substituted aryl groups have an extremely detrimental effect on the reactivity of this reaction probably due to the steric hinder effect. On the other hand, substituents at the quinone methide fragment were tolerated as well, and the corresponding adducts **4j-o** were obtained in good to excellent enantioselectivities (81–89% ee). Notably, alkyl-substituted o-QMs were also demonstrated to be acceptors amenable to the reaction protocol, giving rise to the corresponding products **4p-q** with enantioselectivities up to 75% ee. However, a variety of ortho-aminobenzyl alcohols could not afford the desired product probably due to the low reactivity of aza-o-quinone methide. The absolute configuration of **4l** was determined to be (S, S,S) on the basis of single crystal X-ray diffraction analysis.

Subsequently, we turned our attention to the substrates scope of α-propynyl benzyl alcohols by using o-QM precursor **2 h** as substrate. As shown in Fig. 4, a wide range of α-propynyl benzyl alcohols **3b-i** bearing different substituted aryl groups reacted with **2 h** quite well to form the corresponding poliketide

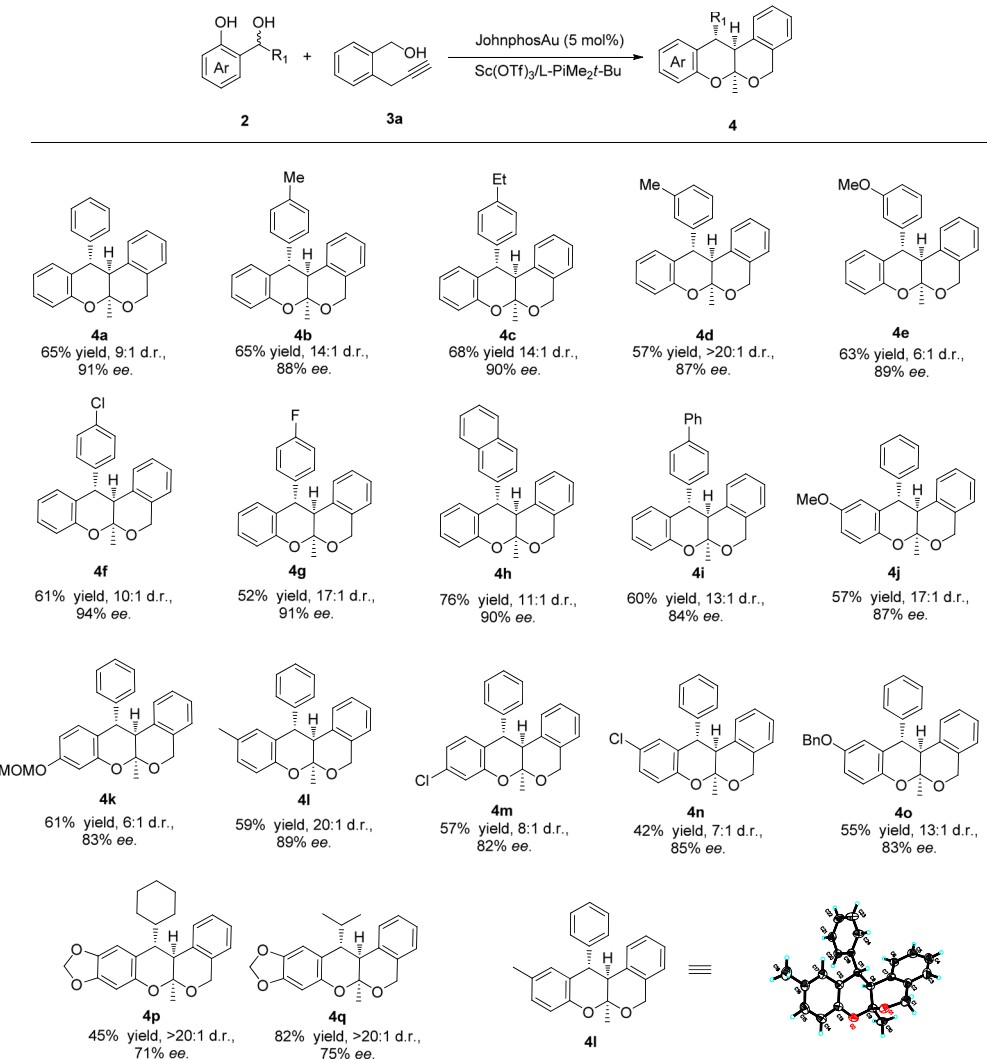

**Fig. 3 Substrates scope of *o*-QMs.** Unless otherwise noted, all reactions were carried out on a 0.2 mmol scale with 1.0 eq precursor of o-QMs **2**, 1.2 eq **3a**, 10 mol% of Sc(OTf)$_3$, 11 mol% of L-PiMe$_2$*t*-Bu and 5 mol% of JohnphosAu in DCM (2.0 mL) under 6 °C. Isolated yield for major diastereomer. *ee* was determined by chiral HPLC. D.r. was detected by crude $^1$H-NMR.

oligomers **4r-4y** in good to moderate yields (43–72%) with excellent *ee* values (89–95%). In general, electron-donating groups furnished the poliketides **4r** and **4s** in higher yields than those with electron-withdrawing substituents **4t-y**, albeit in similarly high *ee* values. Cbz-protected α-propargyl benzyl amine **3j** as well as various alkyl benzyl alcohol derivatives **3k-q** was investigated in this cascade reaction under the optimized reaction conditions. **3j** reacted with **2h** quite well to form the corresponding N-*aza* poliketide oligomer **5a** in 63% yield, 1.5:1 d.r with 22% *ee* for major diastereoisomer and 15% *ee* for minor diastereoisomer. The reactions between **2h** and α-ethynyl benzyl alcohols **3k-l** were carried out with L-PiAd as ligand and chiral spiro products **5b-c** were isolated in 72–76% yield and 52–55% *ee*. 2-(Propynyl)phenol **3m** was also tolerated as well under the standard reaction condition and gave the corresponding chiral spiro product **5d** in 31% yield with 37% *ee*. Moreover, alkyl benzyl alcohol **3o** reacted with **2h** to form the product **5f** by 1,4-addition without annulation in 8 membered ring system likely due to the ring strain. It is worth mentioning that the reaction between **3n** and **2h** was messy toward 7 membered ring system and poliketide oligomer **5e** was isolated in 14% yield with 0% *ee*. Internal alkynes **3p-q** were also demonstrated to be reactants

amenable to the reaction protocol, giving rise to the corresponding products **5g-h** in 39–41% yield with 53–55% *ee*. Hexynols **3s-t** are unavailable substrates in this catalytic system. However, pentynol derivative **3r** reacted quite well with **2h** to form the chiral spiro product **5i** in 52% yield with 66% *ee*.

To evaluate the synthetic potential of this protocol, the cascade reaction of **2h** and **3a** was carried out on a gram scale under optimized reaction conditions with Au(I)/*N,N′*-dioxide-Sc(III) complex. The corresponding product, tetracyclic isochroman poliketide oligomer **4h**, could be obtained with 52% yield, 9:1 d.r. and 90% *ee* (Fig. 5a). In addition, derivatizations of product **4** were performed without loss of enantiopurity of the functionalized products. For example, the controllable selective cleavages of C-O bonds in poliketide **4d** were carried out without any deterioration of enantiopurity (Fig. 5b). In the presence of strong Lewis acid BF$_3$•OEt, **4d** was readily cleaved to the corresponding oxonium ion to furnish bioactive compound **6** which was reported to be an estrogen receptor degrader for treatment of ER + breast cancers with good yield and complete diastereocontrol (Fig. 5b, black part). If TiCl$_4$ was used as strong Lewis acid instead of BF$_3$•OEt, **4d** was readily transformed to another oxonium ion and phenol **7** was obtained (Fig. 5b, blue part). Considering the

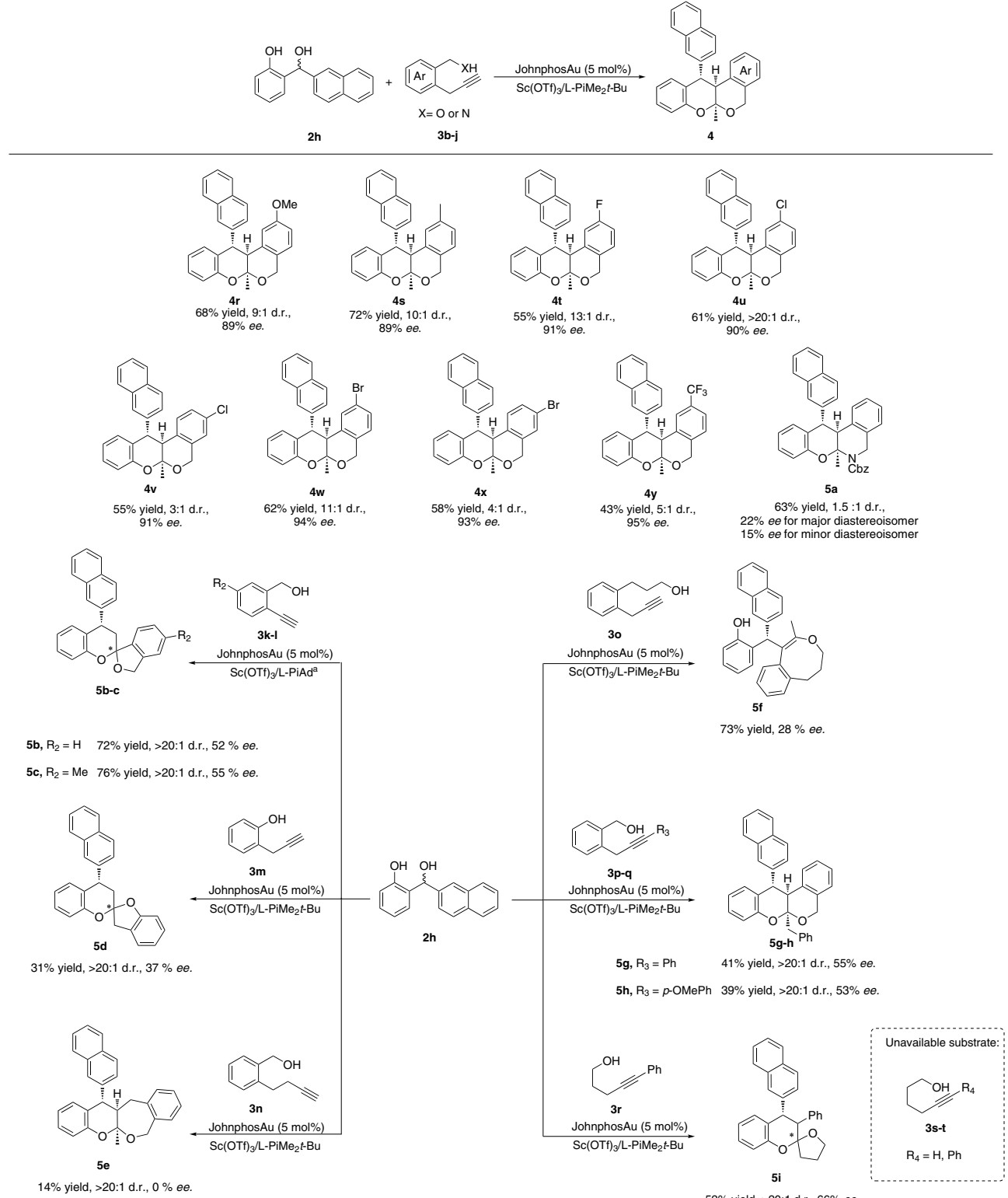

**Fig. 4 Substrates scope of α-propargyl benzyl alcohols.** Unless otherwise noted, all reactions were carried out on a 0.2 mmol scale with 1 eq **2h**, 1.2 eq **3**, 10 mol% of Sc(OTf)$_3$, 11 mol% of L-PiMe$_2$t-Bu and 5 mol% of JohnphosAu in DCM (2.0 mL) under 6 °C. Isolated yield for major diastereomer. ee was determined by chiral HPLC. D.r. was detected by crude $^1$H-NMR. $^a$11 mol% of L-PiAd as ligand.

structure of tetracyclic isochroman natural product, **8r** (R = H) as a precursor of *o*-QM was tested in the asymmetric reaction with **2a**, but no product was obtained. Unexpectedly, TMS substituted *ortho*-hydroxybenzyl alcohol **8s** gave the product **9a** with the

skeleton of natural product directly under standard conditions in 90% *ee* albeit with somewhat eroded yield (Fig. 5c).

As a key step toward establishing the basis for diastereo- and enantiocontrol in this cascade reaction of 1-(hydroxymethyl)-2-

**Fig. 5 The gram scale experiment and transformations of product 4. a** The gram scale experiment. **b** Reduction of product **4d**. **c** Synthesis of analogue of pseudo-natural fungal polyketide **A**.

phenols **2** and α-propargyl benzyl alcohols **3** with the catalysis of Sc(III)/Au(I), we endeavored to determine whether the HDA reaction of in-situ generated isochromene and *ortho*-quinone-methide proceeds via a concerted or stepwise pathway. To address this question, we investigated the carbon isotope effects for the [4 + 2] reaction between **2b** and **3a** using Singleton's method at natural abundance[62,63]. When the reaction was stopped at 71% conversion, the $^{13}C$ ratio of each carbon in the recovered 3-methyl-1H-isochromene **10** to the same carbon in virgin **10** was measured using quantitative $^{13}C$ NMR. As shown in Fig. 6, the only appreciable carbon isotope effect was observed at the 4-C-position of **10**, thus indicating that the HDA cycloaddition might proceed through a stepwise mechanism to form adducts.

In order to investigate the role of the Au-complex in this enantioselective cascade reaction, we investigated the hetero-Diels-Alder reaction between *o*-QMs precursor **2b** and 3-methyl-1H-isochromene **10** with or without the participation of Au-complex (Fig. 7, entry 2 and 3). Two catalytic reactions afforded the tetracyclic isochroman poliketide oligomer **4b** in similar yield, d.r. and *ee* values comparing to the cascade reaction between **2b** and **3a** (Fig. 7, entry 1), which suggests that **10** may be the real active intermediate in the [4+2] cycloaddition and Au-complex may not participate in the control of the enantioselectivity.

Based on the reported X-ray structure of the *N,N'*-dioxide Sc$^{III}$ complex[64] and the absolute configuration of the product **4** as well as the results of control experiments, we proposed a catalytic model for the Au/Sc*catalyzed HDA reaction (Fig. 8). A complex with octahedral geometry as the transition state is formed by coordinating L-PiMe$_2$t-Bu and *o*-QMs to Sc$^{III}$ center. The enol attack takes place from the less hindered *Si* face of the *o*-QMs to form the first chiral center in the benzylic position of *o*-QM skeleton (Fig. 8, 1a *VS* 1b). Subsequent oxygen anion attack to oxonium ion affords the desired optical tetracyclic isochroman from *Si* face, while the *Re* face of the oxonium ion is shielded by the *N,N'*-dioxide (Fig. 8, 1c *VS* 1d).

In summary, we designed and developed the first asymmetric HDA reaction of in-situ generated isochromene and *o*-QM based on insights from the biosynthetic pathway of polyketide oligomer

**Fig. 6 Carbon isotope effects (R/R$_o$) calculated for 3a.** The carbon valued in bold was taken as the internal standard.

natural products. This direct enantioselective cascade reaction was achieved through a Au(I)/Sc(III) bimetallic catalytic strategy, affording a series of chiral tetracyclic isochromans from readily available α-propargyl benzyl alcohols and 2-(hydroxylmethyl) phenols under mild conditions. Importantly, this work demonstrates the potential utility of biomimetic synthesis in the development of reaction and expanding the limitation of substrates comparing to biosynthesis expansion, which encourages us to explore biomimetic catalytic reactions in the future.

## Methods

**General experimental procedure of asymmetric cascade reaction**. To a 10-mL test-tube were sequentially added JohnphosAu (0.010 mmol, 7.7 mg), Sc(OTf)$_3$ (0.020 mmol, 9.8 mg), L-PiMe$_2$t-Bu (0.022 mmol, 14.4 mg) and CH$_2$Cl$_2$ (2.0 mL). The mixture stirred for 15 min. α-Propargyl benzyl alcohol **3** (0.3 mmol) and substrate **2** (0.2 mmol) were added in turn to the solution at 6 °C. The reaction mixture was monitored by TLC. Upon completion, the residual was purified by silica gel flash chromatography (petroleum ether:ethyl acetate, 20:1) to afford the desired product **4**. The racemic examples were prepared by the catalysis of Sc (OTf)$_3$ in r.t.

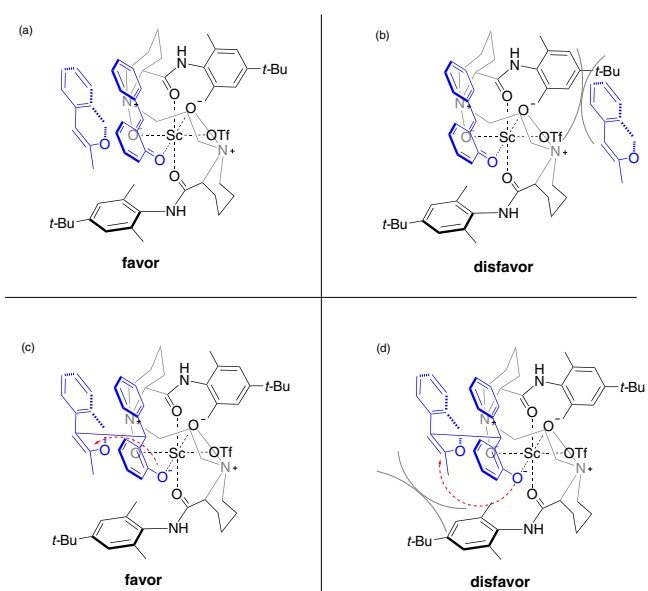

**Fig. 7 Controlled experiments.** Unless otherwise noted, all reactions were carried out on a 0.2 mmol scale with 1 eq **2b**, 1.2 eq **3a** or **10**, 10 mol% of Sc(OTf)$_3$, 11 mol% of L-PiMe$_2$t-Bu and 5 mol% of JohnphosAu in DCM (2.0 mL) under 6 °C.

**Fig. 8 Proposed stereochemical models. a** Favor transition state of first step. **b** Disfavor transition state of first step. **c** Favor transition state of second step. **d** Disfavor transition state of second step.

**Experimental procedure for synthesis of 6**. Under Ar atmosphere, to a 10-mL test-tube were sequentially added **4d** (0.4 mmol, 136.8 mg) (89% *ee*), CH$_2$Cl$_2$ (2.0 mL), Et$_3$SiH (0.4 mmol, 46.4 mg) and BF$_3$·Et$_2$O (0.2 mL) in 0 °C. Upon completion (2 h) the residual was purified by silica gel flash chromatography (petroleum ether: ethyl acetate, 5:1) to afford the desired product **6**.

**Experimental procedure for synthesis of 7**. Under Ar atmosphere, to a 10-mL test-tube were sequentially added **4d** (0.2 mmol, 68.4 mg) (89% *ee*), CH$_2$Cl$_2$ (2.0 mL), Et$_3$SiH (0.2 mmol, 23.2 mg) and TiCl$_4$ (0.1 mL) in 0 °C. Upon completion (2 h) the residual was purified by silica gel flash chromatography (petroleum ether: ethyl acetate, 5:1) to afford the desired product **7**.

## Data availability

Crystallographic data have been deposited in the Cambridge Crystallographic Data. Center under accession number CCDC: 2036557. These data can be obtained free of charge from The Cambridge Crystallographic Data Centre via.

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

## Acknowledgements

We appreciate the National Natural Science Foundation of China (21871254, 21532006, 21472187) and DICP (DICP ZZBS201602).

## Author contributions

X. Lin. performed experiments and prepared the Supplementary Information and manuscript. X. Liu., K.W., and Q.L. participated some discussions and provided some suggestions. Y. L. and C. L. directed the project and helped with modifying the paper.

## Competing interests

The authors declare no competing interests.
