## [Peer Review File · Nature Communications]

Reviewers' Comments:

Reviewer #1:

Remarks to the Author:

Liu and coworkers report the cooperatively catalyzed (4+2)-cycloaddition of in situ generated ortho-quinone methides and enol ethers to produce a broad range of highly valuable tetracyclic isochromans carrying three adjacent stereogenic centers including one tetrasubstituted acetal center. Product yields are typically moderate to good (55-75% for most products), the diastereoselectivity good to very good (10-20:1 d.r.) and the enantioselectivity excellent (ca. 90% ee for most products). As catalysts they employ chiral scandium-N,N-dioxides developed by Feng to promote the dehydration of the starting ortho-hydroxy benzyl alcohols and formation of Lewis acid-bound ortho-quinone methides and a gold-JohnPhos-complex to effect the hydroxylation of the alkyne to the enol ether. The process benefits from the simple and readily available starting materials and the straightforward, one-step generation of complex heterocyclic scaffolds frequently found in polyketide natural products. Moreover, careful mechanistic studies have been undertaken to elucidate the exact reaction mechanism.

This work takes advantage and is a very nice illustration of cooperative catalysis with two catalysts individually activating two substrates in two separate catalytic cycles making possible a highly challenging transformation which otherwise would not have been possible or at least have taken several more steps. In addition, it is the first synthesis of linearly fused chromans instead of the previously achieved synthesis of spirocyclic chromans. Taken together this manuscript is strongly recommended for publication in Nature Communications.

Reviewer #2:

Remarks to the Author:

In this manuscript, the authors realized the construction of tetracyclic isochromans through the biomimetic approach. The HDA reaction between o-QM and isochromene, which are in-situ generated by using Au(I) and Sc(III)-N,N'-dioxide complex respectively, supplies a series of tetracyclic products in good results. These two catalysts show good compatibility in this design. However, the narrowed substrates scope has been presented in this work and the detective of mechanism is not enough as well. We believe the manuscript could be published only if the authors could supply more information as required. The main arguments are listed below:

1. The manuscript needs polishing since there are too many mistakes presented. The color of handwriting should be the same black but there are red characters in the paper.

In line 86, "bronsted" should be "Brønsted";

In line 102, "L-Pit-BuMe2" should be modified to maintain consistency with that in other parts;

In line 103, "CHCl3" should be "CH2Cl2";

In line 117, "ee" should be italic and such careless mistakes should be double checked full text;

In line 179 and 182, "N,N'" should be italic;

In line 207, 213, 218, "°C" should be checked.

...

2. On the other hands, the SI should be carefully checked words by words as well.

In many cases, the ¹H NMR experiments are carried out at 400 MHz but the odd J value are presented which is not possible;

Since the products are chiral white solids, then, the melting points together with the optical rotations must be supplied;

The HRMS data in cases like 4d, 4f, 4g, 4h, 4m, 4q, 4r, 4s, 4v, 4w, 4y, 6 are not convincing enough since the great deviation;

In copies of NMR spectra, the structure of compounds and the serial number of compounds should be fully presented but not part of them. Besides, since the d.r. values are determined by ¹H NMR, they should be clearly exhibited in spectra but you have missed unfortunately. In addition, in case 4u, ¹³C NMR should keep one decimal as others.

In HPLC spectra, the impurities are common in spectra which make us doubt about the accuracy of yields. You are required to pure all compounds and supply newly prepared HPLC spectra. In cases like 4d, 4f, 4m, 4n, 4r, 4w, 5, 6, the racemates are presented not even racemic or wrong data, you need a careful check for all these mistakes.

3. As for substrate scope, only ortho-hydroxybenzyl alcohols and α -propargyl benzyl alcohols have been tested in your work. You are asked to test substrates like ortho-aminobenzyl alcohols and α -propargyl benzyl amines. On the other hand, only six-membered ring systems have been examined, how about 5, 7, 8, or larger ring systems?
4. The mechanism needs more details. By the way, other catalyst systems should be examined as well to show the superiority of your system. Chiral phosphoric acid, chiral Au(I) system, other ligands like Box and so on are required in your work.
5. *Org. Chem. Front.* 2014, 1, 298 ; *Chin. J. Chem.* 2021, 39, 969 and *Aldrichimica Acta* 2020, 53, 3 should be cited.

Reviewer #3:

Remarks to the Author:

This manuscript, by Li, Liu and coworkers, reported an asymmetric hetero-Diels-Alder reaction between in-situ generated isochromene and ortho-quinonemethide via a Au(I)/chiral Sc(III) bimetallic catalytic system. The authors carried out the experimental studies and disclosed that a stepwise pathway and the enantioselectivity of this reaction was controlled by proposed in-situ generated Au(I)-isochromene complex and Sc(III)/N,N'-dioxide complex. The reaction scope was investigated and gave several tetracyclic isochroman frameworks from α -propargyl benzyl alcohols and 2-(hydroxymethyl) phenols.

Suggestions:

- 1) The authors carried out the model reactions using 2-(Hydroxymethyl) phenol (2a) and α -propargyl benzyl alcohol (3a). Scheme 2 and 3 show the reaction scope by changing the substituted groups of two reactants. Actually, these adducts have very similar and specific structures. During the mechanism studies, substrates 7r and 7s (Scheme 4c) were tested. Interestingly, 8a was obtained from 7s. This reviewer suggests the authors think about the practicability of this reaction in current version. More information should be provided to guide the potential utilities by other groups. a) how about the substrates bearing alkyl groups on the benzylic position, b) how about the substrates bearing internal alkynes; c) instead of α -propargyl benzyl alcohols, how about directly using hexynol or related alkynol without benzene ring. The author should provide both the advantages and the limitations of this reaction.
- 2) A stepwise reaction pathway was proposed based on the studies of carbon isotope effects for the [4 + 2] reaction between 2b and 3a. In principle, two diastereomers should be produced during the second bond forming step. This review suggests a detailed study should be performed by direct use of 3-methyl-1H-isochromene (7, Scheme 5) as a reactant. This will give insight about the reaction pathway.
- 3) In figure 1, proposed stereochemical models were given. A crucial question is what's the role of the Au-complex. Is there any interaction between Au with the ligand? Is that necessary to in-situ form the Au-complex? No information was provided.
- 4) The compounds numbering is messy in the manuscript. For examples, 2a, 7, 8 were used twice in different Schemes. The authors should recheck the manuscript carefully and remove these careless errors before submission.

Response to referee 1:

Li, Liu and coworkers report the cooperatively catalyzed (4+2)-cycloaddition of in situ generated ortho-quinone methides and enol ethers to produce a broad range of highly valuable tetracyclic isochromans carrying three adjacent stereogenic centers including one tetrasubstituted acetal center. Product yields are typically moderate to good (55-75% for most products), the diastereoselectivity good to very good (10-20:1 d.r.) and the enantioselectivity excellent (ca. 90% ee for most products). As catalysts they employ chiral scandium-N,N-dioxides developed by Feng to promote the dehydration of the starting ortho-hydroxy benzyl alcohols and formation of Lewis acid-bound ortho-quinone methides and a gold-JohnPhos-complex to effect the hydroxylation of the alkyne to the enol ether. The process benefits from the simple and readily available starting materials and the straightforward, one-step generation of complex heterocyclic scaffolds frequently found in polyketide natural products. Moreover, careful mechanistic studies have been undertaken to elucidate the exact reaction mechanism. This work takes advantage and is a very nice illustration of cooperative catalysis with two catalysts individually activating two substrates in two separate catalytic cycles making possible a highly challenging transformation which otherwise would not have been possible or at least have taken several more steps. In addition, it is the first synthesis of linearly fused chromans instead of the previously achieved synthesis of spirocyclic chromans. Taken together this manuscript is strongly recommended for publication in Nature Communications.

A: We greatly appreciate your highly positive comments.

Response to referee 2:

In this manuscript, the authors realized the construction of tetracyclic isochromans through the biomimetic approach. The HDA reaction between o-QM and isochromene, which are in-situ generated by using Au(I) and Sc(III)-N,N'-dioxide complex respectively, supplies a series of tetracyclic products in good results. These two catalysts show good compatibility in this design. However, the narrowed substrates scope has been presented in this work and the detective of mechanism is not enough as well. We believe the manuscript could be published only if the authors could supply more information as required. The main arguments are listed below:

Q1. The manuscript needs polishing since there are too many mistakes presented. The color of handwriting should be the same black but there are red characters in the paper.

In line 86, "bronsted" should be "Brønsted";

In line 102, "L-Pit-BuMe₂" should be modified to maintain consistency with that in other parts;

In line 103, "CHCl₃" should be "CH₂Cl₂";

In line 117, "ee" should be italic and such careless mistakes should be double checked full text;

In line 179 and 182, "N,N" should be italic;

In line 207, 213, 218, "°C" should be checked.

A1: Thank you for your carefulness in reviewing our manuscript. All mistakes have been corrected.

Q2-1. On the other hands, the SI should be carefully checked words by words as well.

In many cases, the 1H NMR experiments are carried out at 400 MHz but the odd J value are presented which is not possible;

A2-1: Thanks a lot! According to your suggestion, we checked the SI very carefully and revised some mistakes. We also recalculated the *J* values of 400 MHz ¹H NMR by multiplying difference of chemical shift of keeping three decimal places by 400. It needs to note that original *J* values were calculated by keeping four decimal places and rounding to one decimal place.

Q2-2. Since the products are chiral white solids, then, the melting points together with the optical rotations must be supplied;

A2-2: The melting points together with the optical rotations were supplied in the revised Supporting Information.

Q2-3. The HRMS data in cases like 4d, 4f, 4g, 4h, 4m, 4q, 4r, 4s, 4v, 4w, 4y, 6 are not convincing enough since the great deviation;

A2-3: According to your suggestion, HRMS data of **4d, 4f, 4g, 4h, 4m, 4q, 4r, 4s, 4v, 4w, 4y, 6** were remeasured and the deviations between the calculated HRMS data and found data were reduced to lower than one thousandths.

Q2-4. In copies of NMR spectra, the structure of compounds and the serial number of compounds should be fully presented but not part of them. Besides, since the d.r. values are determined by ¹H NMR, they should be clearly exhibited in spectra but you have missed unfortunately. In addition, in case 4u, ¹³C NMR should keep one decimal as others.

A2-4: In the copies of NMR spectra, all of the compounds have been fully presented. All d.r. values have been exhibited in ¹H NMR spectra. Moreover, ¹³C NMR of **4u** has kept one decimal as others.

Q2-5. In HPLC spectra, the impurities are common in spectra which make us doubt about the accuracy of yields. You are required to pure all compounds and supply newly prepared HPLC spectra. In cases like 4d, 4f, 4m, 4n, 4r, 4w, 5, 6, the racemates are presented not even racemic or wrong data, you need a careful check for all these mistakes.

A2-5: We repeated all the experiments which did not supply clean HPLC traces. The products were purified as much as we can. The yields were recalculated and the clean HPLC traces were supplied. Moreover, the racemates of **4d, 4f, 4m, 4n, 4r, 4w, 5, 6** were obtained again and were presented as racemic version in HPLC.

Q3. As for substrate scope, only ortho-hydroxybenzyl alcohols and α -propargyl benzyl alcohols have been tested in your work. You are asked to test substrates like ortho-aminobenzyl alcohols and α -propargyl benzyl amines. On the other hands, only six membered ring system have been examined, how about 5, 7, 8, or larger ring system?

A3: Thank you for your suggestions. According to your suggestions, a variety of *ortho*-aminobenzyl alcohols, Cbz-protected α -propargyl benzyl amine **3j** as well as various alkyl benzyl alcohol derivatives **3k-o** were prepared and investigated in the present catalytic reaction system under the standard reaction conditions. Cbz-protected α -propargyl benzyl amine **3j** can react with **2h** quite well to form the corresponding *aza*-poliketide oligomer **5a** in 63% yield with 22% *ee* for major isomer. Moreover, 5, 7 and 8 membered ring systems were examined. The reactions between **2h** and α -ethynyl benzyl alcohols **3k-l** were carried out with **L-PiAd** as ligand

and chiral spiro products **5b-c** were isolated in 72-76% yield and 52-55% *ee* values in 5 membered ring system. 2-(Propynyl)phenol **3m** was also tolerated as well under the standard reaction conditions and gave the corresponding chiral spiro product **5d** in 31% yield with 37% *ee*. Alkyl benzyl alcohol **3o** reacted quite well with **2h** to form the product **5f** by 1,4-addition without annulation in 8 membered ring system likely due to the ring strain. However, a variety of *ortho*-aminobenzyl alcohols could not afford the desired product probably due to the low reactivity of aza-*o*-QM. It is worth mentioning that the reaction between **3n** and **2h** was messy toward 7 membered ring system and poliketide oligomer **5e** was isolated in only 14% yield with 0% *ee* after purification. The above results were added in the manuscript.

Q4. The mechanism needs more details. By the way, other catalyst systems should be examined as well to show the superiority of your system. Chiral phosphoric acid, chiral Au(I) system, other ligands like Box and so on are required in your work.

A4. In order to investigate the role of the Au-complex in this enantioselective cascade reaction, we investigated that the hetero-Diels-Alder reaction between *o*-QMs precursor **2b** and 3-methyl-1H-isochromene **10** with or without the participation of Au-complex (Scheme 6, entry 2 and 3). Two catalytic reactions afforded the tetracyclic isochroman poliketide oligomer **4b** in similar yield, d.r. and *ee* values comparing to the cascade reaction between **2b** and **3a** (Scheme 6, entry 1), which suggests that **10** may be the real intermediate in the [4+2] cycloaddition and Au-complex may not participate in the control of the enantioselectivity. Based on the reported X-ray structure of the *N,N'*-dioxide Sc^{III} complex and the absolute configuration of the product **4** as well as the result of control experiment, a catalytic model was proposed for the HDA reaction to explain the origin of enantio- and diastereoselectivity of the process (Figure 1).

Moreover, chiral phosphoric acid, chiral Au(I) system, other ligands like Box and so on were examined in this reaction. Chiral phosphoric acids and Sc(III)-Box complexes afforded the racemic product in moderate yield. No product was obtained in chiral Au(I) system. However, chiral acetonitrile-Au(I) system afforded the products in little lower *ee* values (see SI, table S2).

Q5. Org. Chem. Front. 2014, 1, 298; Chin. J. Chem. 2021, 39, 969 and Aldrichimica ACTA 2020, 53, 3 should be cited.

A6. These references have been cited in the revise manuscript (see ref. 53, 59 and 60).

Response to referee 3:

This manuscript, by Li, Liu and coworkers, reported an asymmetric hetero-Diels–Alder reaction between in-situ generated isochromene and ortho-quinonemethide via a Au(I)/chiral Sc(III) bimetallic catalytic system. The authors carried out the experimental studies and disclosed that a stepwise pathway and the enantioselectivity of this reaction was controlled by proposed in-situ generated Au(I)-isochromene complex and Sc(III)/*N,N'*-dioxide complex. The reaction scope was investigated and gave several tetracyclic isochroman frameworks from α -propargyl benzyl alcohols and 2-(hydroxymethyl) phenols.

Q1. a) The authors carried out the model reactions using 2-(Hydroxymethyl) phenol (2a) and α -propargyl benzyl alcohol (3a). Scheme 2 and 3 show the reaction scope by changing the

substituted groups of two reactants. Actually, these adducts have very similar and specific structures. During the mechanism studies, substrates 7r and 7s (Scheme 4c) were tested. Interestingly, 8a was obtained from 7s. This reviewer suggests the authors think about the practicability of this reaction in current version. More information should be provided to guide the potential utilities by other groups. a) how about the substrates bearing alkyl groups on the benzylic position, b) how about the substrates bearing internal alkynes; c) instead of α -propargyl benzyl alcohols, how about directly using hexynol or related alkynol without benzene ring. The author should provide both the advantages and the limitations of this reaction.

A1. Thanks a lot! According to your suggestion, we synthesized *ortho*-hydroxybenzyl alcohols bearing alkyl groups on the benzylic position **2p-q**, internal alkyne **3p-q**, hexynol as well as pentynol derivative and investigated the catalytic activity under the optimized reaction conditions.

a) The substrates bearing alkyl groups on the benzylic position **2p-q** were demonstrated to be acceptors amenable to the reaction protocol, giving rise to the corresponding products **4p-q** in 45-82% yield with enantioselectivities up to 75% *ee* (see scheme 2).

b) Internal alkynes **3p-q** were also demonstrated to be reactants amenable to the reaction protocol, giving rise to the corresponding products **5g-h** in 39-41% yield with 53-55% *ee* (see scheme 3).

c) Hexynols **3s-t** are unavailable substrates in this catalytic system. However, pentynol derivative **3r** reacted quite well with **2h** to form the chiral spiro product **5i** in 52% yield with 66% *ee*. (see scheme 3).

The above results were added in the manuscript.

Q2. A stepwise reaction pathway was proposed based on the studies of carbon isotope effects for the [4 + 2] reaction between 2b and 3a. In principle, two diastereomers should be produced during the second bond forming step. This review suggests a detailed study should be performed by direct using 3-methyl-1H-isochromene (7, Scheme 5) as a reactant. This will give insight about the reaction pathway.

A2. According to your suggestion, we investigated that the hetero-Diels-Alder reactions between *o*-QMs precursor **2b** and 3-methyl-1H-isochromene **10** with or without the participation of Au-complex (scheme 6, entry 2 and 3). Two catalytic reactions afforded the tetracyclic isochroman poliketide oligomer **4b** in similar yield, d.r. and *ee* values comparing to the cascade reaction between **2b** and **3a** (scheme 6, entry 2 and 3 vs entry 1).

Q3. In figure 1, proposed stereochemical models were given. A crucial question is what's role of the Au-complex. Is there any interaction between Au with the ligand? Is that necessary to in-situ form the Au-complex? No information was provided.

A3. Thank you for your suggestion. The result of control experiment in Scheme 6 suggests that Au-complex may not participate in the control of the enantioselectivity and there may be no interaction between the Au-complex and *N,N'*-dioxide ligand. Based on the reported X-ray structure of the *N,N'*-dioxide Sc^{III} complex and the absolute configuration of the product **4** as well as the result of control experiment, a catalytic model was proposed for the HDA reaction to explain the origin of enantio- and diastereoselectivity of the process (Figure 1). The *Re* face of the *o*-QMs is shielded by the neighboring *t*-Bu group of the ligand and the enol attack takes place from the *Si* face of the *o*-QMs to form the first chiral center in benzylic position of *o*-QM skeleton

(Figure 1a VS 1b). Subsequent oxygen anion attack to oxonium ion afford the desired optical tetracyclic isochroman from *Si* face, while the *Re* face of the oxonium ion is shielded by the *N,N'*-dioxide (Figure 1c VS 1d). On the other hands, the Au-complex *in-situ* formed could not transform α -propargyl benzyl alcohol to 3-methyl-1H-isochromene and no reaction occurred (see SI, table S2, entry 9 and 10). Only (acetonitrile)Au-complex could catalyze the reaction.

Q4. The compounds numbering is messy in the manuscript. For examples, 2a, 7, 8 were used twice in different Schemes. The authors should recheck the manuscript carefully and remove these careless errors before submission.

A4. Thanks a lot for your carefulness. We rechecked the manuscript carefully and corrected the messy compound numbers.

Reviewers' Comments:

Reviewer #2:

Remarks to the Author:

The authors have carefully investigated and addressed the concerns that reviewers raised in the peer-review. With plenty of additional experiments, the mechanism of this process could be better explored. In addition, it is glad that all of concerns have been satisfactorily addressed in the revised manuscript. Thus, acceptance of this revised manuscript is now recommended for publication in Nature Communications. Last but not least, Nat. Comm. 2021, 12, 3012 should be cited.

Reviewer #3:

Remarks to the Author:

This reviewer appreciates the authors, who have tried their best to address all of my the questions. I believe the revised version becomes suitable for acceptance by Nature Communications in its current form.

Reviewer #2 (Remarks to the Author):

Q: The authors have carefully investigated and addressed the concerns that reviewers raised in the peer-review. With plenty of additional experiments, the mechanism of this process could be better explored. In addition, it is glad that all of concerns have been satisfactorily addressed in the revised manuscript. Thus, acceptance of this revised manuscript is now recommended for publication in Nature Communications. Last but not least, Nat. Comm. 2021, 12, 3012 should be cited.

A: Thank you for your suggestion. Nat. Comm. 2021, 12, 3012 has been cited in ref 61.

Reviewer #3 (Remarks to the Author):

Q: This reviewer appreciates the authors, who have tried their best to address all of my the questions. I believe the revised version becomes suitable for acceptance by Nature Communications in its current form.

A: Thank you for your comments.